# Patient and caregiver perspectives of select non-communicable diseases in India: A scoping review

Sindhu Nila[1], Eliza Dutta[2], S. S. Prakash[3], Sophy Korula[4], Anu Mary Oommen[5]*

1 KEM Hospital Research Centre, Rasta Peth, Savitribai Phule Pune University, Ganeshkhind, Pune, Maharashtra, India, 2 Indian Institute of Public Health, Shillong, Pasteur Hills, Lawmali, Shillong, Meghalaya, India, 3 Department of Biochemistry, Christian Medical College Vellore, Vellore, Tamil Nadu, India, 4 Department of Paediatrics, Christian Medical College Vellore, Vellore, Tamil Nadu, India, 5 Department of Community Health, Christian Medical College Vellore, Vellore, Tamil Nadu, India

* anuoommen@cmcvellore.ac.in

## Abstract

### Background and objectives

Patient-reported measures of encounters in healthcare settings and consideration of their preferences could provide valuable inputs to improve healthcare quality. Although there are increasing reports of user experiences regarding health care in India in recent times, there is a lack of evidence from Indian healthcare settings on the care provided for patients with chronic diseases.

### Methods

We selected diabetes mellitus and cancer as representatives of two common conditions requiring different care pathways. We conducted a scoping review of studies reporting experiences or preferences of patients/caregivers for these conditions, in PubMed, Global Index Medicus and grey literature, from the year 2000 onwards. Both published and emergent themes were derived from the data and summarised as a narrative synthesis.

### Results

Of 95 included studies (49 diabetes, 46 cancer), 73% (65) were exclusively quantitative surveys, 79% included only patients (75), and 59.5% (44) were conducted in government centres. Studies were concentrated in a few states in India, with the underrepresentation of vulnerable population groups and representative studies. There was a lack of standardised tools and comprehensive approaches for assessing experiences and preferences of patients and caregivers, concerning diabetes and cancers in India. The commonest type of care assessed was therapeutic (74), with 14 cancer studies on diagnosis and nine on palliative care. Repeated visits to crowded centres, drug refill issues, unavailability of specific services in government facilities, and expensive private care characterised diabetes care, while cancer care involved delayed diagnosis and treatment, communication, and pain management issues.

**Data Availability Statement:** All relevant data are within the manuscript and its Supporting information files.

**Funding:** The author(s) received no specific funding for this work.

**Competing interests:** The authors have declared that no competing interests exist.

## Conclusions

There is a need for robust approaches and standardised tools to measure responsiveness of the healthcare system to patient needs, across geographical and population subgroups in India. Health system reforms are needed to improve access to high-quality care for treatment and palliation of cancer and management of chronic diseases such as diabetes.

## Introduction

Responsiveness to the needs and expectations of patients and their families, is one of the three goals of the health system, in addition to improving health outcomes and fairness in financial contribution [1]. A responsive health system considers respect for dignity, confidentiality, and autonomy of patients, as well as orientation to the needs of clients [1]. India was ranked in the third decile, among 195 countries, in the Healthcare Access and Quality Index (Global Burden of Diseases, 2016), with wide disparities between states in the HAQ index, based on levels of amenable mortality (deaths which should not have occurred) and other outcomes that could be prevented by health care [2]. While the low HAQ index points to the need for improving health systems especially for NCDs [2], understanding subjective experiences and preferences of consumers is also necessary to be responsive to their needs and improve the quality of healthcare services. In order to assess health system performance both the average level of responsiveness, and the levels of responsiveness to different socio-demographic groups, need to be assessed [1].

While high-income countries have long-standing systems and guidelines for quality assessment of health services, such as the Hospital Consumer Assessment of Healthcare Providers and Systems (HCAHPS) in the United States [3] and the patient experience framework for the National Health Services in the United Kingdom [4], this concept is relatively new in low and lower-middle-income countries. In recent years, hospitals and other health centres in India have been seeking accreditation from the National Accreditation Board for Hospitals & Healthcare Providers (NABH) [5], which includes regular patient feedback, in addition to other standard procedures, to improve patient outcomes and satisfaction. However, although NABH accreditation is displayed by facilities, there is no system of public reporting of patient feedback similar to the HCAHPS [3].

Non-Communicable Diseases (NCDs) are the leading cause of morbidity, disability, mortality, and economic losses globally. While there are reviews and studies on patient experiences of conditions such as tuberculosis, maternal and childhood care [6, 7], there is no systematic literature review on perspectives of people living with NCDs in India. There is a need to understand the perspectives of these affected persons, to ensure that the health system is responsive to their needs.

Our scoping review aimed to identify and map the evidence documenting patients' and caregivers' experiences and preferences, pertaining to care received for cancer and diabetes mellitus, from diverse healthcare settings in India, using these two diseases as exemplars of NCDs. We chose these diseases mainly due to their high disease burden in India. Diabetes mellitus showed the highest increase in Disability-Adjusted Life Year (DALY) rates (39%) among major NCDs in India, between 1990 and 2016 [8], while the contribution of cancers to total DALYs doubled during this period [9]. These diseases also represent two widely different pathways for NCD care, diabetes mellitus being an example of a disease requiring multiple encounters with the health system over a lifetime, while cancer represents a chronic disease with a shorter, often pain-filled journey, from diagnosis to final outcome. Diabetes and cancers are

also two of the major NCDs covered by the national NCD programme (earlier called the National Programme for Prevention and Control of Cancer, Diabetes, Cardiovascular Diseases and Stroke, NPCDCS) from 2012. Thus, while many other NCDs are equally important from a public health perspective, including hypertension, cardiovascular diseases, stroke, chronic respiratory disorders, and mental health disorders, we chose diabetes mellitus and cancer as case studies, to highlight the challenges patients and their families encounter while seeking care for common NCDs.

We chose to conduct a scoping review, due to the broad nature of our questions and to understand the extent of available evidence related to diabetes and cancer care with the following questions:

1. What is the availability or gap in evidence regarding experiences of patients and caregivers utilising care in healthcare facilities in India?

2. What is the evidence regarding what patients and their caregivers prefer/expect, while utilising healthcare in Indian healthcare settings?

## Materials and methods

We framed this scoping review according to the Preferred Reporting Items for Systematic Reviews and Meta-Analysis extension for Scoping Reviews (PRISMA-ScR) (S1 Table) [10], and the Joanna Briggs Institute (JBI) scoping review guidelines [11]. We pre-registered the protocol at the Open Science Framework website of the Centre for Open Science (https://osf.io/cwgz3, doi: 10.17605/OSF.IO/CWGZ3).

The databases searched were PubMed and Global Index Medicus (GIM), supplemented by grey literature from Google Scholar and Shodhganga (an Indian database on dissertations and theses, https://shodhganga.inflibnet.ac.in/). We searched references of included articles and conducted additional hand searching, with the last search conducted on 9th August 2022. The detailed search strategies, including MeSH terms and keywords are given in S1 File.

The search terms were based on the "Population-Concept-Context" (PCC) principle [11].

Population: Patients with either type 1 or type 2 diabetes mellitus or cancer, of any age or sex, and/or their caregivers.

Concept: Experiences, satisfaction and/or preferences of patients/caregivers.

Context: Healthcare received in Indian health facilities, in either private or government sectors, at any level of care (primary/secondary/tertiary), for patients with cancer or diabetes mellitus, from the year 2000 onwards.

### Operational definitions of outcomes/concepts

**Patient / caregiver experiences.** Studies that reported patients' or caregivers' interactions with the healthcare system, assessing the quality of care received at facilities, with self-reported measures of satisfaction, documenting both positive and negative experiences.

**Patient preferences or expectations.** Studies reported on choices expressed by patients and/or caregivers, values that were considered imperative, and expectations related to facility-based care.

The HCAHPS survey domains [3] were used to categorise experiences, with combining of some domains and addition of inductively coded themes:

- Communication (communication with doctors /nurses/ other staff, communication regarding medication, responsiveness of staff),

- Transition of care (delays in diagnosis and treatment due to referrals),

- Hospital environment (cleanliness, quietness, other organisational aspects of facilities such as overcrowding),

- Pain management,

- Discharge information,

- Overall rating of hospital and recommendation of the hospital (overall treatment satisfaction) [3].

The World Health Organization (WHO) framework for domains of responsiveness was also applied (respect for persons and client orientation) [1]. Patient/caregiver experience was categorised on whether the article mainly reported positive, negative, or mixed experiences.

Search terms were generated through brainstorming, piloted in the selected databases, and finalised using an iterative process. AA first developed search strategies for each database independently, and then further improved them through collaborative discussions within our team and by seeking advice from experts in the field (QMED Knowledge foundation, https://www.qmed.ngo/). We conducted two rounds of pilot testing on the search strategy and inclusion/exclusion criteria and made slight modifications after each round.

## Study eligibility

We included all studies that focused on experiences or preferences of patients and/or caregivers, concerning healthcare received at facilities in India, published from the year 2000 onwards. As English is the primary language for scientific publications in India, we restricted the search to English language articles.

We excluded studies without original data (e.g., expert opinions, guidelines), reviews, studies on gestational diabetes mellitus, and those focusing on patient-reported outcomes that were focused solely on clinical, epidemiological, or costing data.

## Screening of articles

Two teams of two reviewers each were formed for diabetes (SN, SK) and cancer (SSP, ED), with a fifth author functioning as an independent reviewer (AMO), for unresolved conflicts. The articles retrieved from each database were imported to the DistillerSR software (Evidence Partners Incorporated; Ottawa, Canada, https://www.evidencepartners.com/), for both screening of articles and final data extraction. Two reviewers independently screened titles and abstracts according to the eligibility criteria (Level 1), followed by full texts of eligible studies (Level 2). Whenever there were disagreements between team authors regarding eligibility, we would discuss and reach a consensus or consult an independent author (AMO), who was not involved in Level 1 and 2 screening. If necessary, the team sought advice from an expert. EE checked a 5% random sample of screened articles in Level 2 to check quality and to ensure consistency in deciding eligibility.

## Data extraction

We used two extraction forms, standardised through pilot testing, to extract information from selected full texts. Form 1 (S2 Table) was used to extract study characteristics including design, setting, level of care (primary/ secondary/ tertiary facility or community-based), and participant demographics. Form 2 (S3 Table) contained information on key findings related to experiences and preferences, stage of care (e.g., screening, diagnosis, treatment, palliation,

survivorship), as well as types of experience. Relevant quotes from qualitative studies were also extracted. The teams independently extracted data for diabetes and cancer. AMO reviewed the extracted forms for validity and consistency of responses and corrected any errors with inputs from the team.

## Data analysis

The extracted data from the selected studies were analysed by descriptive statistics, disaggregated by disease group. We employed both deductive coding using predefined codes from the HCAHPS framework [3] and the WHO patient responsiveness framework [1], and inductive coding. Coding was done independently for diabetes (SN) and cancer studies (AMO), using NVIVO (Release 1.7, QSR International) and Microsoft Excel. We did not perform any risk of bias estimation or assessment of the quality of studies.

## Results

Of the 1825 records identified from different databases, we screened 275 full-text articles, as shown in a PRISMA flow diagram, Fig 1.

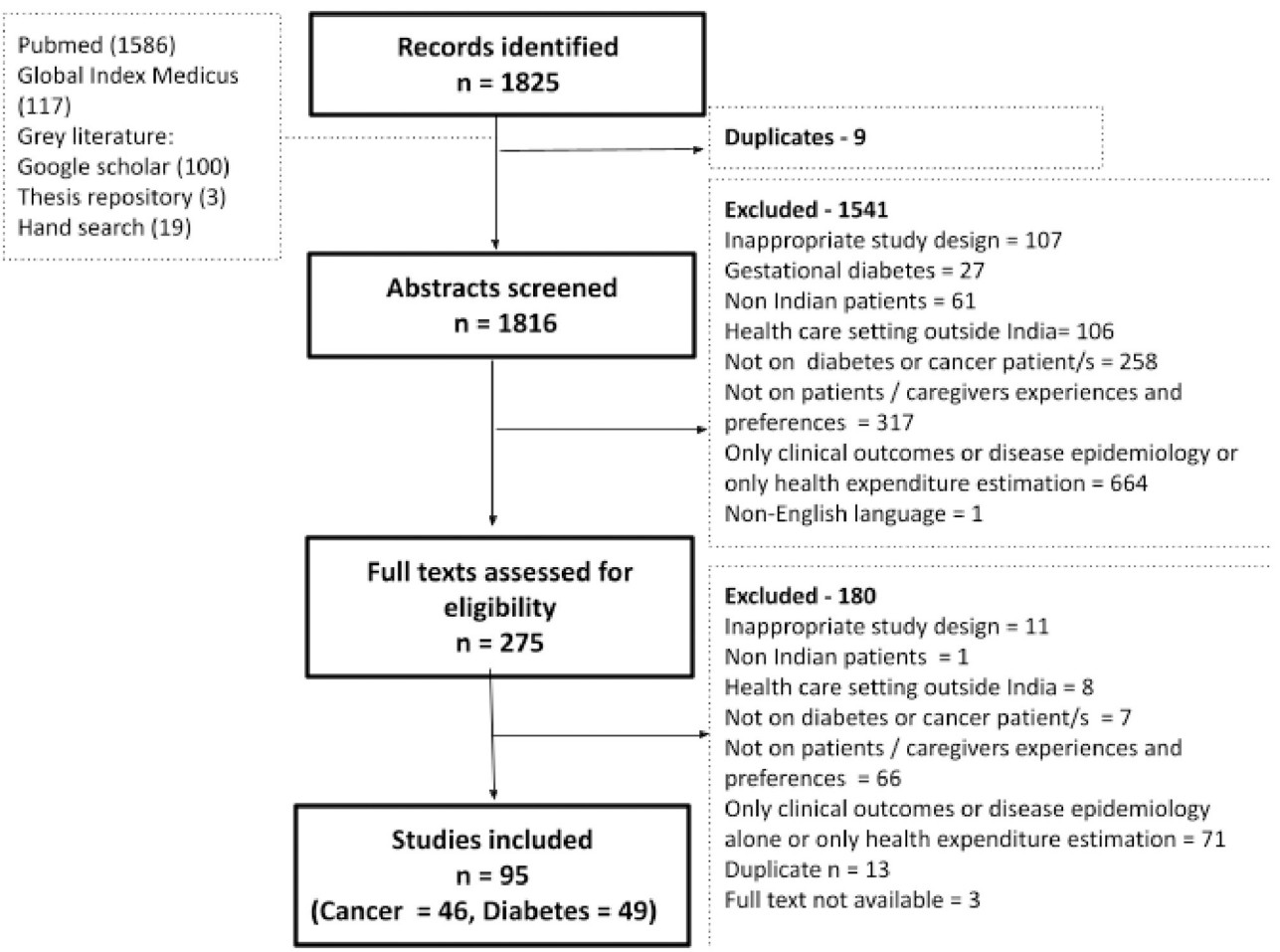

**Fig 1. PRISMA flow diagram for user experiences and preferences for cancer and diabetes—A scoping review.**

We included 95 unique studies (46 cancer, 49 diabetes) after full text screening. The complete list of the 95 studies included in the analysis is in S4 and S5 Tables, along with key study characteristics, setting, year of publication, core strengths, and limitations.

## Characteristics of included studies and key outcomes

Of the 95 included studies, the southern states of Karnataka, Tamil Nadu, Kerala, Andhra Pradesh, Telangana, and Pondicherry contributed the most studies for both diabetes (26 out of 49, 53·1%) and cancer (18 out of 46, 39·1%), Table 1. Study sites from northern and western regions included Gujarat, Rajasthan, Uttar Pradesh, Uttarakhand, Chandigarh, and Delhi; studies from central India included Maharashtra and Odisha, while the only state with any studies from the northeast was Assam (two cancer studies). There was one nationally representative study for cancer and three multi—regional studies on diabetes. Most studies (n = 66, 69.5%) used a cross-sectional survey design, of which 36 (54.5%, 27 diabetes, nine cancer) involved either random selection or all eligible participants during the study period, while the remaining followed convenience sampling or had unclear description of sampling methods. Of five interventional studies for diabetes, four were randomised controlled trials, while the two interventional cancer studies were non-randomised before-and-after interventions.

Most cancer studies (n = 43, 93·5%) were hospital-based, of which most were in tertiary hospitals (n = 42, 97·7%), while 38·8% (n = 19) diabetes studies were community-based surveys, Tables 1 and 2.

Of the 19 community-based studies related to diabetes, five (26·3%) were exclusively rural, nine (47·4%) urban, and one included both rural and urban settings (5·3%). Of the urban diabetes-related studies, five were conducted in urban slums.

**Table 1. Descriptive characteristics of included studies and settings.**

| Variables | Categories | Cancer n out of 46 (%) | Diabetes n out of 49 (%) | Total n out of 95 (%) |
|---|---|---|---|---|
| **Region of the country** | Multiregional | 1 (2·2) | 2 (4·1) | 4 (4.2) |
| | North, West | 9 (19·6) | 15 (30·6) | 24 (25.3) |
| | Central | 16 (34·8) | 5 (10·2) | 21(22.1) |
| | Northeast | 2 (4·3) | 0 (0) | 2 (2.1) |
| | South | 18 (39·1) | 27 (55·1) | 44 (46·3) |
| **Design** | Observational, survey | 29 (63·0) | 37 (73·5) | 66 (69·5) |
| | Qualitative | 8 (17·4) | 2 (4·1) | 10 (10·5) |
| | Mixed methods | 5 (10·9) | 5 (12·2) | 11 (11·6) |
| | Case report | 2 (4·3) | 0 (0) | 2 (2·1) |
| | Experimental | 2 (4·3) | 5 (10·2) | 7 (7·4) |
| **Year of study*** | 2000–2010 | 7 (15·2) | 9 (18·4) | 16 (16·8) |
| | 2011–2021 | 39 (84·8) | 40 (81·6) | 79 (83·2) |
| **Location** | Hospital/ facility | 43 (93·5) | 31 (63·2) [#] | 74 (77·9) |
| | Community–rural | 0 (0) | 4 (8·2) | 4 (4·3) |
| | Community–urban | 1 (2·2) | 9 (16·3) | 9 (9·5) |
| | Community–mixed | 1 (2.2) | 1 (2·0) | 3 (3·2) |
| | Community–unknown | 1 (2.2) | 4 (8·2) | 5 (5·3) |

*year in which study was conducted or year of publication if there was no mention of study year

[#] one study had both hospital based and community components. Further details given in S4 and S5 Tables.

**Table 2. Descriptive characteristics of hospital/ clinic-based studies.**

| Variables | Categories | Cancer<br>n out of 43 (%) | Diabetes<br>n out of 31 (%) | Total<br>n out of 74 (%) |
|---|---|---|---|---|
| **Level of care** | Primary care | 0 | 2 (6·5) | 2 (2·7) |
| | Tertiary | 42 (97·7) | 21 (67·7) | 63 (85·1) |
| | Mixed sites | 0 | 1 (3·2) | 1 (1·4) |
| | Not mentioned | 1 (2·3) | 7 (22·6) | 8 (10·8) |
| **Setting** | Inpatients | 7 (16·3) | 1 (3·2) | 8 (1·8) |
| | Outpatients | 26 (60·5) | 24 (77·4) | 50 (67·6) |
| | Mixed sites in hospital | 7 (16·3) | 3 (9·7) | 10 (13·5) |
| | Telemedicine | 0 | 2 (6·5) | 2 (2·7) |
| | Not mentioned | 3 (6·9) | 1 (3·2) | 4 (5·4) |
| **Provider type** | Government | 30 (69·8) | 13 (48·4) | 43 (58·1) |
| | Private | 10 (23·3) | 13 (38·7) | 23 (31·1) |
| | Charitable | 3 (6·9) | 1 (3·2) | 4 (5·4) |
| | Government and private | 0 | 2 (6·5) | 2 (2·7) |
| | Not mentioned | 0 | 2 (12·9) | 4 (5·4) |

While most studies were outpatient-based, two out of 49 studies (4·1%) on diabetes were related to telemedicine. Out of the 43 hospital-based cancer studies, the majority (n = 30, 69·8%) were done by government institutions, compared to 15 out of 31 studies (48·4%) for diabetes, Table 2. Caregivers were more likely to be interviewed for cancer studies (n = 17, 36·9%) than for diabetes (n = 3, 6·1%, parents of children with type 1 diabetes = 2, family members of patients = 1), Table 3. Out of the 46 cancer studies, nine (19·6%) were exclusively on female patients (Table 3), four of which were focused on breast cancer, three on cervical cancer and two on both cancers.

The outcomes assessed by most studies were related to experiences of patients or caregivers (n = 37 cancer, n = 48 diabetes), with fewer studies reporting preferences (n = 20 cancer, n = 13 diabetes), Table 4.

All 49 diabetes-related studies were related to aspects of therapeutic management, with six studies additionally assessing experiences related to screening for complications [12–17]. While most cancer studies were related to diagnosis (n = 15) or treatment (n = 24), nine focused on palliative care and end of life [18–26], two on survivorship [27, 28] and one on the assessment of the informed consent process for a trial [29], Table 4. Of the 49 studies on diabetes mellitus, only two focused on type 1 diabetes with both interviewing caregivers of affected children.

## Reported experiences during care

The domains of experience assessed are shown in Table 4, with the classification of experiences as positive, negative or mixed, in Table 5. Communication-related experiences and preferences were the most commonly reported outcomes for cancer (n = 18, 39·1%), while both communication (n = 22, 44·9%) and overall satisfaction were equally assessed for diabetes (n = 22, 44.9%). Delays with diagnosis and treatment constituted another common theme for cancer studies (n = 15, 32·6%). Other cancer-related issues included pain management (n = 5, 10·9%) and stigma faced due to hospital visits for cancer treatment (n = 3). Examination of key outcomes assessed across the years revealed that studies assessing perspectives on palliative care,

**Table 3. Characteristics of participants in included studies.**

| Variables | Categories | Cancer n out of 46 (%) | Diabetes n out of 49 (%) | Total n out of 95 (%) |
|---|---|---|---|---|
| Index patient type | Children only | 6 (13·0) | 2 (4·1) | 8 (8·4) |
| | Adults only | 38 (82·6) | 47 (95·9) | 85 (89·5) |
| | Both | 2 (4·3) | 0 | 2 (2·1) |
| Participants | Patients only | 29 (63·0) | 46 (93·8) | 75 (79·0) |
| | Caregivers only | 8 (17·4) | 1 (2·0) | 9 (9·5) |
| | Both | 9 (19·6) | 2 (4·1) | 11 (11·6) |
| Number of participants | < 100 | 17 (36·9) | 14 (28·6) | 31 (32·6) |
| | 100 or more | 29 (63·0) | 35 (71·4) | 64 (67·4) |
| Gender of interviewed/ index patients | Male patients only | 2 (4·3) | 7 (14·3) | 9 (9·5) |
| | Female patients only | 9 (19·6) | 3 (6·1) | 12 (12·6) |
| | Both male, female | 25 (54·3) | 32 (65·3) | 57 (60·0) |
| | Gender data missing | 10[#] (21·7) | 7 (14·3) | 17 (17·9) |
| Gender of interviewed caregivers | Male caregivers only | 1 (2·2) | 0 | 1 (1·1) |
| | Female caregivers only | 2 (4·3) | 1 (2·0) | 3 (3·2) |
| | Both male, female | 5 (10·9) | 0 | 5 (5·3) |
| | Gender data missing | 9 (19·6) | 2 (4·1) | 11 (11·6) |

[#] one study was on breast cancer

**Table 4. Outcomes measured in included studies.**

| Variables | Categories | Cancer n out of 46 (%) | Diabetes n out of 49 (%) | Total n out of 95 (%) |
|---|---|---|---|---|
| Type of care* | Screening | 0 | 6[#] (12·2) | 6 (6·3) |
| | Diagnostic | 14 (30·4) | 0 | 14 (14·7) |
| | Treatment/ management | 25 (54·3) | 49 (100·0) | 74 (77·9) |
| | Palliative and end of life care | 9 (19·6) | 0 | 9 (9·5) |
| | Others | 3 (6·5) (survivorship 2, trial consent 1) | 0 | 3 (3·2) |
| Outcomes | Experiences only | 26 (56·5) | 36 (73·5) | 62 (65·3) |
| | Preferences only | 9 (19·6) | 1 (2·0) | 9 (9·5) |
| | Both | 11 (23·9) | 12 (24·5) | 24 (25·3) |
| Type of experiences described* | Communication, staff responsiveness | 18 (39·1) | 22 (46·9) | 40 (42·1) |
| | Pain management | 5 (10·9) | 0 | 5 (5·3) |
| | Transition of care/ delays in diagnosis, referrals, treatment initiation | 15 (32·6) | 0 | 15 (15·8) |
| | Hospital environment, services | 6 (13·0) | 9 (18·4) | 15 (15·8) |
| | Overall rating/ treatment satisfaction | 8 (17·4) | 22 (44·9) | 30 (31·6) |
| | Others | 4 (8·7) (stigma due to hospital visits 3, alternative medicine 2, access issues leading to default 1) | 8 (16·3) (accessibility 1, telemedicine 2, alternative medicine 5) | 12 (12·6) |

*more than one category possible for each study

[#] screening for complications (retinopathy, neuropathy, nephropathy)

**Table 5. Domain wise reporting of positive, negative, or mixed experiences.**

| Domain of experience | Cancer n (%) | | | | Diabetes n (%) | | | |
|---|---|---|---|---|---|---|---|---|
| | Mainly positive | Mainly negative | Mixed | N | Mainly positive | Mainly negative | Mixed | N |
| Communication | 7 (38·9) | 8 (44·4) | 3 (16·7) | 18 | 3 (13·0) | 10 (47·8) | 9 (39·1) | 22 |
| Pain management | 1 (20·0) | 3 (60·0) | 1 (20·0) | 5 | 0 | 0 | 0 | 0 |
| Transition of care/ delays in diagnosis, referrals, treatment initiation | 0 | 15 (100·0) | 0 | 15 | 0 | 0 | 0 | 0 |
| Hospital environment / organizational aspects | 1 (16·7) | 4 (66·6) | 1 (16·7) | 6 | 1 (12·1) | 6 (66·7) | 2 (22·2) | 9 |
| Overall facility rating/ treatment satisfaction | 3 (37·5) | 3 (37·5) | 2 (25·0) | 8 | 4 (18·2) | 8 (36·4) | 9 (40·9) | 22 |
| Telemedicine | 0 | 0 | 0 | 0 | 1 (50·0) | 0 | 1 (50·0) | 2 |
| Accessibility of facility | 0 | 1 (100·0) | 0 | 1 | 0 | 0 | 1 (100·0) | 1 |
| Alternative medicine use* | 0 | 0 | 0 | 3# | 5* (80·0) | 0 | 0 | 5 |
| Stigma | 0 | 3 (100·0) | 0 | 3 | 0 | 0 | 0 | 0 |

*positive towards alternative medicine but negative towards allopathy

#3 studies mentioned patients trying traditional healers but did not mention if the experience was positive/negative

diagnostic disclosure and decision control preferences for cancer had increased in the past ten years (S5 Table).

Domains of responsiveness (WHO framework) [1] reported by included studies are shown in Fig 2.

Overall, mainly negative experiences were reported by 66.7% (16/24) of cancer studies conducted in government hospitals, compared to only 30.0% (3/10) in private or charitable hospitals. Similarly, 53.8% (7/13) of diabetes-related studies in government hospitals reported mainly negative experiences compared to 23.1% (3/13) in private centres. Analysis by geographical region revealed that 83.3% (5/6) studies that reported cancer-related experiences from north India reported predominantly negative experiences, compared to 75.9% (8/12) from central India, 100% (2) from north-east and 50.0% from south India (8/16). Similarly, for diabetes-related experiences, 53.3% (8/15) of studies from north India reported predominantly negative experiences, compared to 20.0% (1/5) from central India and 46.2% from south India (12/26).

**Communication experiences.** Communication was reported in terms of clarity, respectful communication and communication related to diagnosis and decisions (Fig 2).

Of the 18 studies which reported communication experiences for cancer, seven reported mainly positive experiences, e.g., medical fraternity being a source of strength, appreciation of the style of communication and information provided by doctors. Eight studies reported mainly negative experiences, such as lack of communication to patients about cancer diagnosis, exclusion from decision making [19, 30], dissatisfaction with consultation time, privacy concerns and interruptions during consultations [31–33]. One study reported a quote from a patient with cancer as *"doctor doesn't look at the face; writes a long list of investigation; I don't believe these doctors"* [32] while a caregiver of a patient referred for palliative care said *"nobody tells correctly what has happened, what has to be done"* [24].

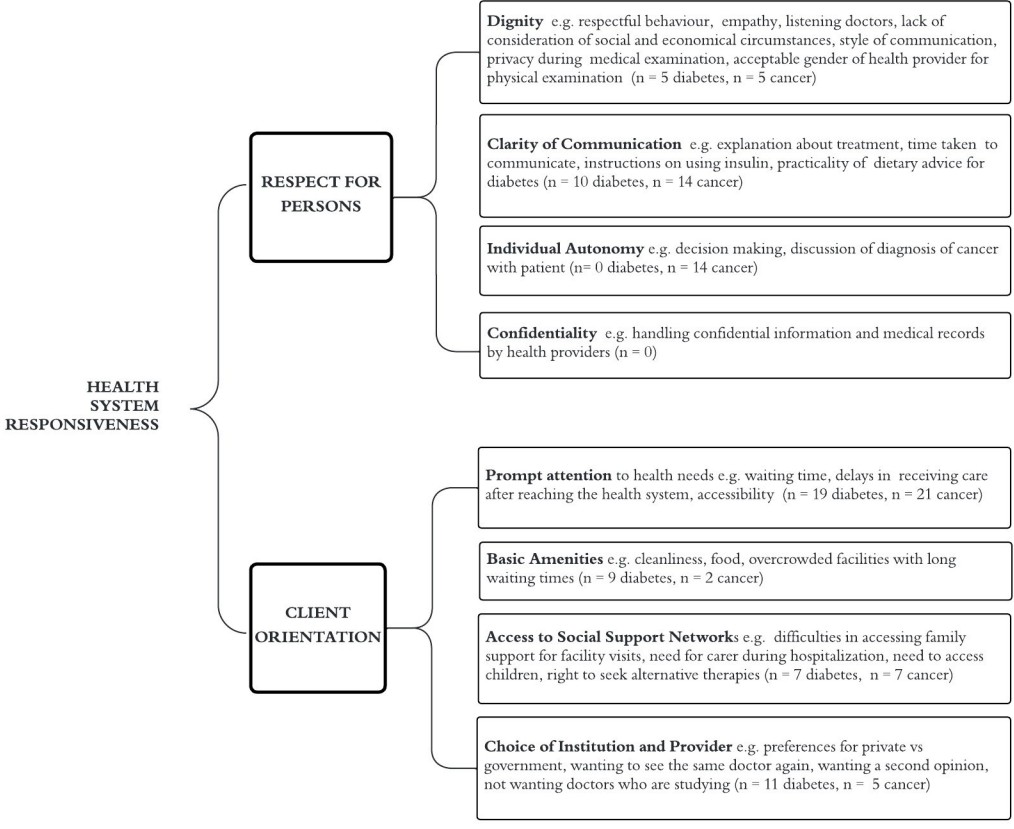

**Fig 2. Application of World Health Organisation responsiveness domains to review findings.**

Highlighting inadequate diabetes-related communication for diet, one study reported: *"Only if he tells us not to eat this, or to eat only that, we will know about it. But if he himself doesn't tell us, what will we know? We are uneducated, so we will simply sit quietly"* [34]. Another aspect of poor communication in the context of diabetes, was with respect to lack of advice regarding screening for complications. One study from a tertiary centre reported that only 28%, 35% and 55% of patients had ever been advised screening for eye, heart, and kidney problems, respectively [35]. Of the 22 studies assessing communication-related experiences for diabetes, only three reported mainly positive experiences with communication, which included satisfaction with ophthalmology services [12], acceptability of text messaging for reminders about treatment [36], and a mobile health communication application [37].

**Delays in management.** Fifteen cancer studies reported health system delays and difficulties in the transition of care due to delayed/missed cancer diagnosis by the initial health provider (diagnostic delay), delays in initiating treatment (treatment delay) and referral delays. For breast cancer, one study from Delhi reported a higher average number of consultations before diagnosis for rural women (3.9), compared to urban women (2.4), among those attending a tertiary government centre [38]. Another study found that 22% of women with cervical cancer seen at a cancer hospital mentioned that not being diagnosed by general practitioners was the cause of delayed treatment [39].

**Hospital environment and organizational aspects.** For cancer-related care, experiences related to basic amenities and services were mostly negative (n = 6), such as dissatisfaction with cleanliness, waiting areas, laboratory services, hospital diet, overcrowding and long

waiting times in clinics. Seven diabetes-related studies described similar reasons for dissatisfaction, such as overcrowding and long queues, mainly in public sector facilities. Other reasons for dissatisfaction included the need for frequent visits for refilling diabetic medication: *"We have to keep coming, stand in the queue only for 2-week medicines"*, and poor outpatient medical records, *"Whatever it is, it is available in this slip, where do I find the old slips now"* [40]. Lack of specific investigations and drugs, and unavailability of doctors and insulin at primary care facilities also caused dissatisfaction about the quality of medical care. These experiences led to patients seeking treatment from private health providers, incurring significant out-of-pocket expenses, even after being seen in government facilities [40, 41]. One study on diabetes from a government medical college from south India reported positive feedback about waiting time and that satisfaction with waiting for treatment/investigations was associated with better quality of life [42].

**Pain management.**   Pain management for cancer was reported by five studies, which highlighted several issues, including needing to go to the hospital for opioids, hesitancy of doctors in prescribing opioids, and uncontrolled pain even with hospitalization [25]. One study reported mixed responses to palliative pain management and highlighted both a limited understanding of palliative care as merely for pain management, as well as an appreciation of such services as exemplified by this quote from a caregiver: *"This treatment is for the body pain so that she must not experience the body pain. But this also is helping because pain is the main thing now"* [24]. Another study which used a validated tool (FAMCARE-2) to measure satisfaction with advanced cancer care, reported high satisfaction of family caregivers with inpatient palliative care at a tertiary government centre in south India, especially in terms of pain and other physical symptoms [26].

**Overall satisfaction with the facility or treatment.**   Eight cancer studies reported overall experiences regarding satisfaction with treatment, with three reporting high satisfaction with quality of care. Das et al. assessed retrospective regret in parents of children who had died of cancer, finding that only 30% of these parents had chosen hospitals as end-of-life care locations for their children, with 62·5% of them regretting their decision [25]. Other issues included abandoning treatment and feeling angry over treatment [43].

For diabetes, 22 studies reported various measures of overall satisfaction with treatment or facility. Some studies used specific scales like quality of life measurements with subcomponents assessing treatment satisfaction (n = 6), Diabetes Treatment Satisfaction Questionnaire [44] and Patient Assessment of Chronic Illness Care (PACIC) [45].

Dissatisfaction with allopathic treatment led to patients choosing complementary or alternative medicines (CAM) (n = 5 diabetes, n = 3 cancer) [44, 46–49].

Health system factors leading to poor compliance for diabetes management (n = 8) included dissatisfaction with symptom relief, distance, drug refills being provided only for two weeks, inadequate instructions, treatment costs, waiting times and non-availability of certain services [34, 37, 45, 50–54].

## Preferences for care

Among the 46 cancer studies, 20 assessed preferences related to care (patients alone = 11 studies, caregivers alone = 3 studies, both = 6 studies). Eight studies assessed communication and information needs, such as treatment details, expenses, and illness disclosure, with patients expressing a desire to be informed about more than what their relatives wanted them to know [18, 27, 55–59]. Studies which reported patients' preferences regarding management explored expectations from palliative care, acceptable risk of drug toxicity, preferences for surgery to extend life, decision control preferences, wanting access to family members during treatment

and patient criteria for choosing hospitals. One study among patients with pancreatic cancer described that satisfaction with services/organization was a preferred outcome for > 80% of patients [22]. Two studies reported preferences of breast/ cervical cancer survivors on preferred places of death and information needs [27, 28].

Among the studies on diabetes, 12 assessed preferences (patients alone = 11, caregivers alone = 1). Three studies assessed preferences related to screening for diabetic retinopathy [12, 15, 17]. One study highlighted that while patients were happy with teleophthalmology services, they were not a replacement for physical consults due to a lack of direct communication and a short time for consultation [12]. Preference for complementary and alternative medicine (CAM) over allopathic treatment for diabetes was primarily due to fewer side effects, low costs, accessibility, and desire for early benefits [46, 48].

Patients also preferred investigations to be available at the same facility as treatment for diabetes, as represented by the following quote:

> *"At XXX [a government center], they say that I have to get [my] blood checked at some other place [private laboratory] and take the report to them to get the medication. I do not want it in that manner. . .. If you want to help poor people, all the facilities should be there at one place."*
>
> *(55-year-old-woman)* [34].

Other diabetes-related studies assessed preferences for private facilities, teleophthalmology, health communication, and longer refills of medication [17, 34, 41, 57].

## Discussion

Understanding the experiences and preferences of patients and their caregivers as the end-users of services is vital for a responsive health system. Service quality assessment has been defined in the SERQUAL (service quality) measurement framework as the gap between expectations and experiences with current services, which helps measure the functional quality of health care (as opposed to technical quality related to clinical practices and outcomes) [60].

We found that the distribution of studies was not uniform across the country, with half the diabetes-related studies and more than a third of the cancer studies done in south India. The northeastern part of India, and states with the least socioeconomic development in central and northern India, were underrepresented. Vulnerable populations, including transgender persons, people with disabilities, and migrant and tribal populations, were underrepresented. The finding that most studies (n = 79, 82·1%) were conducted in the last ten years (2011–2021), indicates increasing interest in understanding patients' perspectives. Although limited, qualitative studies included in this review provided valuable insights and need to be employed more frequently to improve the quality of care.

We also found a need for standardised methods used across India for assessing patient experiences. Many studies did not assess patient experiences as a primary objective, but reported these as part of other findings, such as compliance with treatment, quality of life, complementary and alternative medicine use, and awareness and attitude surveys. We chose a scoping review because of the breadth of the outcomes and the types of reports we assessed. As the indicators used were highly heterogeneous, a meta-analysis with the current evidence was impossible. The quality of the evidence was also doubtful, with poor reporting of definitions and results, and poor generalisability of most studies (S4 and S5 Tables), although we did not use a formal tool for evaluating the quality.

We limited this review to diabetes mellitus and cancer as two examples of NCDs, for which patients and caregivers need frequent encounters with the health system, albeit with different care pathways. Both diabetes mellitus and cancer contribute to major disease burden [8, 9, 61], poor quality of life, and economic burden on families and health systems. Cancer is a painful disease, often rapidly fatal if untreated, posing multiple challenges for families, including obtaining and accepting a diagnosis, and accessing expensive treatment that is available only in higher centres in India. Diabetes on the other hand, is more likely to require prolonged, life-long treatment from multidisciplinary teams, with a need for repeated facility visits, access to affordable drugs and follow-up care. Studies included in this review highlight these distinctive aspects of both diseases, with cancer studies focusing on patient/ caregiver perspectives and challenges in obtaining a diagnosis, palliative and/or therapeutic care. In contrast, diabetes-related studies mainly focused on patient experiences related to difficulties in routine outpatient care. Although our study did not include articles reporting the experiences/preferences of patients with hypertension alone, which is also a significant risk factor for cardiovascular disease, we believe the issues faced by such patients are similar to those highlighted in this review for diabetes.

Patient feedback is often mandated for the accreditation of hospitals by many national accreditation boards [62], including the NABH in India, which specifies patient satisfaction as an indicator of quality for quality improvement and accreditation [5]. The studies included in our review did not include social media ratings of consumers, or unpublished reports of passively collected feedback collected by many hospitals for quality improvement. While this implies that the studies included in this review, which have all employed active feedback, are more representative than reports from passive feedback, studies involving passive feedback are also needed, as they have been shown to provide more information on negative experiences and suggestions for improvement [63].

Similarly, in our review, there was a dearth of user feedback from facilities outside research settings, with the domination of tertiary care hospital-based studies. India's flagship programme, the National Health Mission, includes a component of community-based monitoring of health facilities including public hearings, interviews of patients and community members [64]. These can be an excellent platform for user feedback to improve healthcare delivery in public health systems. However, this feedback is not available in the public domain.

Communication-related experiences were the most studied domain of patient experience, with evidence highlighting poor experiences, including poor communication with patients regarding the diagnosis and treatment of cancer, lack of empathetic attitudes, and poor communication regarding diet and screening for complications of diabetes. While privacy during examination and issues with caregivers not wanting patients to be informed of their diagnosis were reported in a few cancer studies, no studies reported experiences or preferences regarding confidentiality of medical records or information revealed to providers, which could be due to low expectations of the public regarding such issues. Given the plans for nationwide digitisation of health records, preparing the health system to deal with confidentiality issues will be essential. Training of medical personnel in lifestyle education, communication, and counselling, needs greater emphasis in medical and nursing education, keeping in mind language, social and educational barriers [65].

The experiences of patients and caregivers affect their choices, expectations, and preferences for future encounters with the health system, in addition to reported satisfaction. Compared to the number of studies focussed on experiences, there were fewer studies focused purely on assessing preferences, such as patient-reported outcome measures or preferences regarding treatment. Further, a lack of clarity on differentiating concepts of experiences, preferences, and expectations (what people expect to happen during interactions) was evident

[66]. Given that prior expectations influence patient experiences, reported satisfaction and subsequent expectations, studies applying frameworks such as developed by Lakin et al. [66], can help future research on patient expectations. One area in which we noted emerging evidence was in the context of palliative care preferences, indicating increasing involvement of patients and families in palliative care. The higher number of such studies after 2010–12, could be related to the launch of the NPCDCS programme in India.

The preferences of patients and caregivers reported in the included studies, such as information needs, empathetic listening, decreased waiting time, and decision control preferences were similar to results from other countries [67]. Preference for alternative forms of treatment for diabetes demonstrates the current dissatisfaction with allopathic medication and treatment, and also indicates the higher responsiveness of these usually smaller facilities that do not pose the same challenges faced in allopathic facilities [68, 69]. Preference for private care for treatment of diabetes, although costly, reflects the need to improve the quality and organisation of care in the primary healthcare system.

These preferences indicate the need for revamping the delivery of services in India's public healthcare facilities to reinforce trust, in line with the National Health Policy 2017 objectives, which seeks to provide universal health coverage [70].

Diagnostic delays experienced by patients with cancer were similar to patient experiences for chronic respiratory disease in India described by Kane et al., painting a picture of two stages in health care experiences: an urgent search for a cure that drives patients to multiple care providers who are often private, followed by acceptance, when the need is for relief of symptoms, rather than cure [71]. The initial stage very often leads to poverty and highlights the lack of reliable, accessible primary care in many states, which was also confirmed by our review. Another review of cancer care during the recent pandemic also highlighted diagnostic and treatment delays due to the pandemic, with impact on communication between patients and providers, exacerbating psychological ill health [72].

## Recommendations

Health system reforms are needed to improve patient-provider interaction at the service provider, managerial and policymaker levels to improve both satisfaction and clinical outcomes [73]. There is a need to develop standardised, contextualised approaches for measuring responsiveness to be used across public and private facilities in developing countries, building on existing tools and frameworks for health system responsiveness [69]. Ideally public reporting of feedback surveys, as done for HCAHPS [3], could potentially enable end users to compare facilities and increase accountability of health systems. Tools could be generic, such as the Patient Assessment of Chronic Illness Care [74], HCAHPS [3], and the WHO patient responsiveness framework [1], or disease specific. Our review had fewer studies from private facilities than government facilities, which was concerning, as they contribute significantly to out-of-pocket expenses in India. Measurement of responsiveness across all types of healthcare facilities is necessary to obtain a more accurate and fairer picture. Although the small number of studies from various regions was a limitation that precluded comparison of experiences by region, studies from north and western India tended to report more negative experiences than south and central India, which needs further exploration.

Although patient experiences and expectations are related to patient satisfaction, additional factors, such as health status and factors external to the health system influence satisfaction [75]. Thus, while the reported studies mostly portray negative experiences of the interactions of patients and caregivers with health services, the underlying reasons for dissatisfaction may not all be related to the health system. There is a need for broader societal interventions and

policy measures to ensure ease of access to quality health care. Interventions to improve communication, patient and caregiver involvement in decision-making, organization of care and referral pathways, and quality of amenities, must be complemented by improved social support systems, health literacy, and accessibility to healthcare facilities [1].

## Conclusions

Our scoping review maps the available literature and gaps in evidence of patient and caregiver perspectives regarding care for diabetes mellitus and cancer in India. The absence of robust approaches, and underrepresentation of subgroups of populations and geographical regions, point to the need for a comprehensive strategy of evaluating users' experiences of the health system to inform improvements in the health system in India. The existing evidence highlights challenges faced in the diagnosis and initiation of treatment for cancer, as well as the continuation of care for diabetes, including poorly equipped government facilities and referral systems. These issues force patients to turn from government to private facilities, seek alternative options and point to India's pressing need for health system reforms.

## Supporting information

**S1 Table. Preferred Reporting Items for Systematic reviews and Meta-Analyses extension for Scoping Reviews (PRISMA-ScR) checklist.**
(DOCX)

**S2 Table. Data extraction form for study characteristics.**
(DOCX)

**S3 Table. Data extraction form for key findings.**
(DOCX)

**S4 Table. Descriptive characteristics of included diabetes-related studies (n = 49) in chronological order.**
(DOCX)

**S5 Table. Descriptive characteristics of included cancer-related studies (n = 46) in chronological order.**
(DOCX)

**S1 File. Search strategy.**
(DOCX)

## Acknowledgments

This article has been written as part of the research for the Lancet Citizens' Commission on Reimagining India's Health System. The views expressed are those of the author(s) and not necessarily those of the Lancet Citizens' Commission or its partners.

We are grateful to the Christian Medical College (CMC) Vellore, for mentorship and resources for this work.

We are thankful to Professor Gagandeep Kang, Christian Medical College Vellore, for her role in conceptualisation and critical inputs at each step of the study, and Professor Sumit Kane, the University of Melbourne, for his critical revision.

We thank Dr. Vasumati and the QMED team for their guidance in testing and finalizing our search strategy, and Dr. Dipanwita Sengupta for her guidance.

## Author Contributions

**Conceptualization:** Sindhu Nila, Eliza Dutta, S. S. Prakash, Sophy Korula, Anu Mary Oommen.

**Data curation:** Sindhu Nila, Anu Mary Oommen.

**Formal analysis:** Sindhu Nila, Anu Mary Oommen.

**Investigation:** Sindhu Nila, Eliza Dutta, S. S. Prakash, Sophy Korula, Anu Mary Oommen.

**Methodology:** Sindhu Nila, Eliza Dutta, S. S. Prakash, Sophy Korula, Anu Mary Oommen.

**Project administration:** Anu Mary Oommen.

**Supervision:** Anu Mary Oommen.

**Writing – original draft:** Sindhu Nila, Eliza Dutta, Anu Mary Oommen.

**Writing – review & editing:** S. S. Prakash, Sophy Korula.

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
