## [Decision Letter · Decision Letter 0]

10 Jul 2023

PONE-D-23-11835Patient and caregiver perspectives of select non-communicable diseases in India: a scoping reviewPLOS ONE

Dear Dr. Oommen,

Thank you for submitting your manuscript to PLOS ONE. After careful consideration, we feel that it has merit but does not fully meet PLOS ONE’s publication criteria as it currently stands. Therefore, we invite you to submit a revised version of the manuscript that addresses the points raised by reviewers during the review process.

We look forward to receiving your revised manuscript.

Kind regards,

Hariom Kumar Solanki, M.D.

Academic Editor

PLOS ONE

Journal Requirements:

Reviewers' comments:

Reviewer's Responses to Questions

**Comments to the Author**

1. Is the manuscript technically sound, and do the data support the conclusions?

Reviewer #1: Partly

Reviewer #2: Yes

2. Has the statistical analysis been performed appropriately and rigorously? 

Reviewer #1: N/A

Reviewer #2: N/A

3. Have the authors made all data underlying the findings in their manuscript fully available?

Reviewer #1: Yes

Reviewer #2: Yes

4. Is the manuscript presented in an intelligible fashion and written in standard English?

Reviewer #1: No

Reviewer #2: Yes

5. Review Comments to the Author

Reviewer #1: The findings will enhance understanding of the current healthcare services in mentioned area. It will help to improved on the area that need to be improvised. Further enhancement on the format and language need to be done to increase the quality of the paper.

Reviewer #2: An interesting topic for review within the Indian health care context. However, some issues need to be addressed before making the manuscript suitable for publication.

1. Please give appropriate rationale for selecting the two non communicable diseases, viz. Diabetes and Cancer, and not including Hypertension for the review, which is a risk factor for cardiovascular deaths in India.

2. Appreciate the authors that they have mentioned how the keywords for search was selected. Kindly mention the keywords and MeSH terms used in various permutations for conducting the literature search.

3. Please include a table reflecting the general study characteristics including the study setting, year of study and strengths and limitations of each study included in the scoping review, which would enable the readers to better put the findings and also observe any time trend specific changes in patient and caregiver perspective, since the infrastructure, services under NPCDCS (Now NPNCD) and guidelines have been dynamic and evolving.

4. It would be good to see geographically stratified analysis done for patient satisfaction or at least include a write up regarding this in the discussion section, since in India, health is a state subject and wide patient and caregiver differentials are observed in different settings.

6. PLOS authors have the option to publish the peer review history of their article (what does this mean?). If published, this will include your full peer review and any attached files.

Reviewer #1: No

Reviewer #2: **Yes: **Aftab Ahmad

---

## [Author Response · Author response to Decision Letter 0]

14 Aug 2023

Changes made

2. Please include captions for your Supporting Information files at the end of your manuscript, and update any in-text citations to match accordingly. Please see our Supporting Information guidelines for more information: http://journals.plos.org/plosone/s/supporting-information

Added captions

Comment Response Changes made if any

1. Is the manuscript technically sound, and do the data support the conclusions?

Reviewer #1: Partly

Reviewer #2: Yes 

Response: We hope that the specific changes made in response to other comments done would have addressed Reviewer 1’s rating. We also broke up long sentences to improve readability.

2. Has the statistical analysis been performed appropriately and rigorously?

Reviewer #1: N/A

Reviewer #2: N/A 

Response: N/A

3. Have the authors made all data underlying the findings in their manuscript fully available?

Reviewer #1: Yes

Reviewer #2: Yes 

Response: No change asked for

4. Is the manuscript presented in an intelligible fashion and written in standard English?

Reviewer #1: No

Reviewer #2: Yes 

Response: We have revised the language and format; please let us know if you have any further specific suggestions for improvement .

Grammatical corrections were done using grammarly.com (throughout the manuscript, highlighted as tracked changes) and formatting changes were done throughout, according to PLOS One requirements which was also pointed out to us by the Editor. 

References are also placed in square brackets and heading formatted according to PLOS One guidelines.

Tables also have now been placed in the text as indicated in the guidelines.

Double spacing done; headings formatted.

Reviewer #1: The findings will enhance understanding of the current healthcare services in mentioned area. It will help to improved on the area that need to be improvised. Further enhancement on the format and language need to be done to increase the quality of the paper 

Response as above to previous question.

Reviewer #2: 

1. Please give appropriate rationale for selecting the two non communicable diseases, viz. Diabetes and Cancer, and not including Hypertension for the review, which is a risk factor for cardiovascular deaths in India. 

Reponse: 

Yes, we have deliberately chosen only two kinds of common NCDs as it would have taken the review even longer to do and lengthier to read if we included more conditions. Both DM and cancer are common NCDs, and also represent diseases with multiple complications and aspects of treatment with two different types of care pathways, so we chose those as examples of NCDs. We have now added further explanations in limitations based on this concern, acknowledging the importance of hypertension and other common NCDs. Introduction page 4, para 3

Discussion, page 22, para 3

2. Appreciate the authors that they have mentioned how the keywords for search was selected. Kindly mention the keywords and MeSH terms used in various permutations for conducting the literature search. Response: The various combinations of search terms are provided in Supplementary table S2 for all the databases used.

This was too long for the methods section and hence is in a supplementary file table. Mentioned in methods that search terms are given in supplementary File S1: page 6, lines 1-2.

Supplementary File S1 gives both the MeSH terms and keyword permutations.

We hope the supplementary file is accessible to the reviewers.

3. Please include a table reflecting the general study characteristics including the study setting, year of study and strengths and limitations of each study included in the scoping review, which would enable the readers to better put the findings and also observe any time trend specific changes in patient and caregiver perspective, since the infrastructure, services under NPCDCS (Now NPNCD) and guidelines have been dynamic and evolving. 

Since there are nearly a 100 studies this table is given as a set of two supplementary tables S4 and S5. We have added the missing components suggested by Reviewer 2.

We have arranged in chronological order so that readers can view these in table format, and also see changes over time, although such comparative assessment is difficult due to wide heterogeneity in methods and results between studies. 

We have added comments in results and discussion on differences in study outcomes over time. Supplementary Tables S4 and S5 (added more descriptives, strengths and limitations)

Results:

Page 9, para 1,2

Page 15, para 2

Page 17, para 1

Discussion: page 24, para 2

4. It would be good to see geographically stratified analysis done for patient satisfaction or at least include a write up regarding this in the discussion section, since in India, health is a state subject and wide patient and caregiver differentials are observed in different settings. 

Response: 

We have done a sub analysis as suggested, by looking at % with predominantly negative experiences (either overall poor satisfaction or other negative experiences related to communication, delays etc.), based on geographical regions (as already classified in results under ‘characteristics of included studies and key outcomes’). This sub analysis has now been mentioned in results and discussion. Results: page 7, para 2

(classification of regions as given in Table 1 and Results page 9, last para)

Discussion page 25, last para. 

Page 26, para 1

---

## [Decision Letter · Decision Letter 1]

2 Nov 2023

PONE-D-23-11835R1Patient and caregiver perspectives of select non-communicable diseases in India: a scoping reviewPLOS ONE

Dear Dr. Oommen, 

Thank you for submitting your manuscript to PLOS ONE. After careful consideration, we feel that it has merit but does not fully meet PLOS ONE’s publication criteria as it currently stands. Therefore, we invite you to submit a revised version of the manuscript that addresses the points raised during the review process.

We look forward to receiving your revised manuscript.

Kind regards,

Hariom Kumar Solanki, M.D.

Academic Editor

PLOS ONE

Journal Requirements:

**Additional Editor Comments:**

Please address the issues raised by one of the reviewer's comments especially the about the citation error and that some of the statements which would be better in methods section than in the introduction section.

Reviewers' comments:

Reviewer's Responses to Questions

**Comments to the Author**

1. If the authors have adequately addressed your comments raised in a previous round of review and you feel that this manuscript is now acceptable for publication, you may indicate that here to bypass the “Comments to the Author” section, enter your conflict of interest statement in the “Confidential to Editor” section, and submit your "Accept" recommendation.

Reviewer #1: All comments have been addressed

Reviewer #2: All comments have been addressed

Reviewer #3: (No Response)

2. Is the manuscript technically sound, and do the data support the conclusions?

Reviewer #1: Yes

Reviewer #2: Yes

Reviewer #3: Partly

3. Has the statistical analysis been performed appropriately and rigorously? 

Reviewer #1: N/A

Reviewer #2: N/A

Reviewer #3: N/A

4. Have the authors made all data underlying the findings in their manuscript fully available?

Reviewer #1: Yes

Reviewer #2: Yes

Reviewer #3: Yes

5. Is the manuscript presented in an intelligible fashion and written in standard English?

Reviewer #1: Yes

Reviewer #2: Yes

Reviewer #3: No

6. Review Comments to the Author

Reviewer #1: (No Response)

Reviewer #2: Hereby thank the author for taking note of all the comments suggested in the review and making suitable changes in the manuscript.

Reviewer #3: Introduction: The introduction section of the article is very limited and need to be elobarated more, I didn't observe the chronological sequence of presenting the data related to the subject matter, some paragraphs within this section are supposed to be within the method section. Citation errors observed.

Methods: Citation errors within the method.

Concepts and context: Not explained.

Discussion: Talked about more about study limitations in here which not suppose to be, the comparison with other studies is not well organized. Mentioned conclussions and recommendations in you discussion section.

English: The language of the article need to be higher standard.

7. PLOS authors have the option to publish the peer review history of their article (what does this mean?). If published, this will include your full peer review and any attached files.

Reviewer #1: No

Reviewer #2: **Yes: **AFTAB AHMAD

Reviewer #3: No

---

## [Author Response · Author response to Decision Letter 1]

26 Nov 2023

Reviewer #3:

1. Introduction: 

The introduction section of the article is very limited and need to be elobarated more, 

I didn't observe the chronological sequence of presenting the data related to the subject matter, 

some paragraphs within this section are supposed to be within the method section. 

Response: 

We have added a few more lines in the intro.

The current intro is in the following sequence:

• Responsiveness of health systems and India’s performance in the global health quality assessment study, 

• Need for assessing experiences and preferences,

• Lack of systems for assessing quality of health facilities in India unlike some other countries,

• Burden of NCDs and the lack of reviews on NCD care,

• Rationale of choosing diabetes and cancer, 

• Research questions (objectives)

We felt that all of the above should remain in Intro and not methods.

We would welcome suggestions from the Editor/Reviewer 3 about which specific paragraphs currently in introduction we could move to methods.

Changes made in:

Intro (page 3) para 1, lines 3,4, 6-9

Page 4: para 2, lines 10-12

2. Citation errors observed.

Methods: Citation errors within the method.

Response: We thank the reviewer for identifying the citation errors, for which we apologize. Citation errors have been checked and corrected. 

One reference (Ref 1) had a wrong title while others had formatting errors.

There are no retracted articles in the references (response to Editor).

Changes made:

Reference errors corrected throughout.

3. Concepts and context: Not explained.

Response: 

In response to the reviewer’s comments, we have rearranged the section on definitions of outcomes (which are the concept definitions) to come immediately after the PCC para and hope that this has improved the readability.

Context: modifications made

Changes made:

Rearranged methods with operational definitions to come in the PCC section

Page 6, line 11 (context)

4. Discussion: 

Talked about more about study limitations in here which not suppose to be, the comparison with other studies is not well organized. 

Mentioned conclussions and recommendations in you discussion section.

Response:

As there are multiple outcomes and studies are very heterogenous, comparison to other studies is given within each outcome heading and not as a single separate paragraph in the Discussion.

As other points were already discussed under each outcome heading, we have mainly discussed limitations and implications of findings in the discussion.

Recommendations are mentioned as a separate paragraph, but perhaps previously was not clear to the reader as there was no heading (added now).

We have added one more recommendation related to public reporting of quality assessment.

Conclusions are in the last para.

(We apologize that we were not sure what this comment meant as it was not clear if the comment refers to the fact that we have already mentioned recommendations and conclusions or have not mentioned them).

Changes:

Discussion:

Recommendations- page 25, last line and page 26, first line

5. English: The language of the article need to be higher standard.

Response:

We have revised the language again and also asked a native English speaker to review, and extensively corrected based on suggestions given.

Changes made:

Tracked changes throughout the manuscript.

Grammatical corrections previously done using grammarly.com

---

## [Decision Letter · Decision Letter 2]

18 Dec 2023

Patient and caregiver perspectives of select non-communicable diseases in India: a scoping review

PONE-D-23-11835R2

Dear Dr. Oommen,

We’re pleased to inform you that your manuscript has been judged scientifically suitable for publication and will be formally accepted for publication once it meets all outstanding technical requirements.

Kind regards,

Hariom Kumar Solanki, M.D.

Academic Editor

PLOS ONE

Additional Editor Comments (optional):

Reviewers' comments:

Reviewer's Responses to Questions

**Comments to the Author**

1. If the authors have adequately addressed your comments raised in a previous round of review and you feel that this manuscript is now acceptable for publication, you may indicate that here to bypass the “Comments to the Author” section, enter your conflict of interest statement in the “Confidential to Editor” section, and submit your "Accept" recommendation.

Reviewer #3: (No Response)

2. Is the manuscript technically sound, and do the data support the conclusions?

Reviewer #3: Partly

3. Has the statistical analysis been performed appropriately and rigorously? 

Reviewer #3: Yes

4. Have the authors made all data underlying the findings in their manuscript fully available?

Reviewer #3: Yes

5. Is the manuscript presented in an intelligible fashion and written in standard English?

Reviewer #3: Yes

6. Review Comments to the Author

Reviewer #3: The sequence of the article has improved as well as the language, but I would like to add only last two comments regarding the conclusion, here it need to be eloborated little bit by connecting it to your results, you have made a good points in the results but just generalized in the conclussion. The other point is very important and it is the discussions, you have mentioned here different parts of your methods and limitations, you dont have to have explain how you choose your methods or what kind of limitations you hade in you dissussion, just compare your result with other litteratures that is all.

7. PLOS authors have the option to publish the peer review history of their article (what does this mean?). If published, this will include your full peer review and any attached files.

Reviewer #3: No

---

## [Editor Report · Acceptance letter]

27 Dec 2023

PONE-D-23-11835R2 

PLOS ONE

Dear Dr. Oommen, 

I'm pleased to inform you that your manuscript has been deemed suitable for publication in PLOS ONE. Congratulations! Your manuscript is now being handed over to our production team.

Kind regards, 

on behalf of

Dr. Hariom Kumar Solanki 

Academic Editor

PLOS ONE